# Failed Mimicry: The Thai Government's Attempts to Combat Labor Trafficking Using Perpetrators' Means

**Naparat Kranrattanasuit * and Yanuar Sumarlan ***

Institute of Human Rights and Peace Studies, Mahidol University, Salaya Campus,
Nakhon Pathom 73170, Thailand
* Correspondence: naparat.kra@mahidol.edu (N.K.); yanuar.sum@mahidol.edu (Y.S.)

**Abstract:** (1) Background: This research paper examines the prevention measures, i.e., the application of technologies such as those abused by "traffickers", used by government and non-government agencies to combat "internal trafficking" in Samut Sakhon province. The authors review numerous research papers and documents at international and national levels. (2) Methods: the authors use in-depth interviews to relate the anti-internal trafficking measures of the government and non-government agencies. (3) The findings show that these government and non-government agencies have attempted to combat "inter-border" trafficking and internal trafficking. However, limited information and communication gaps in the application of IT-based technology and other media for communication have caused unsatisfactory preventive results and responses against such phenomena. (4) Some findings point to the limited success of an NGO (the Labor Protection Network) whose leader decided to recruit Burmese- and Lao-speaking staff to reach out to potential and actual victims among Burmese and Laotian people. (5) The authors suggest that government agencies learned from this failure and then collaborated more with non-government and migrant worker organizations to provide sufficient information and efficient communication channels to ensure migrant workers' safety in Thailand's territory.

**Keywords:** technology-facilitated internal trafficking; preventive measures; safe migration; migrant workers; Thailand

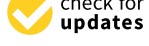



## 1. Introduction

Trafficking in persons has caused a deprivation of fundamental human rights and freedom and national and human security in many developing countries, including Thailand. This crime also leads to a loss of human capital and remittances and tax revenues that generate economic and social insecurity (May 2017). Elements of human trafficking include the "recruitment, transportation, harboring or receipt of persons, by means of coercion, abduction, deception or abuse of power or vulnerability, for the purpose of exploitation", with exploitation including, at a minimum, "sexual exploitation, forced labor, slavery or slavery-like practices" (UNODC 2019). Many studies have revealed that this crime has been expedited by information and communication technologies through mobile phones, the internet, or other online social networks. In particular, traffickers have used "cyber-space" for selling victims and services and attracting customers (Sykiotou 2017).

Although Samut Sakhon (or Mahachai) is just a tiny province (545,216 acres) located at Tha Chin Estuary near Bangkok, the province has a slogan of "Fishery Town, Industrial Community, Agricultural Field, Historic Site". Migrant workers from Myanmar, Cambodia, and Laos in Samut Sakhon province, Thailand have encountered technology-facilitated labor exploitation because this location is one of the most attractive destinations for migrant jobseekers (U.S. Department of State 2019; May 2017). There is abundant evidence that some migrant workers have been trafficked into fishing vessels and shrimp-processing factories. This phenomenon occurs because Samut Sakhon has various sea product-processing



facilities as an important port city through which water-based trade and business traffic occurs (Samut Sakhon Governor's Office n.d.). The phenomenon dubbed "internal trafficking" (Brayley and Cockbain 2014) is very common in Samut Sakhon. With around 300,000–400,000 migrant workers out of Samut Sakhon's population of 450,000 (Sandar 2011), the province sees around 42 percent of these workers in seafood production, 20 percent in services, and 8 percent in agriculture (Areeprachakun 2020, pp. 36–37). Sandar (2011, p. 7) reported that most migrant workers come from Mon State in Myanmar through the Thai border by walking through the forest. This method is much cheaper than the cost of migration from Rakhine State in Myanmar, which might be higher than THB35,000 (Sandar 2011, p. 7). Migrants, who are often sold by a broker into indentured labor on fishing boats until their debt is paid, work for one or two years without a salary from the Thai owners. When they are free from debts, these migrant laborers can work in land-based factories. This phenomenon fits the definition of "internal trafficking", through which foreigners who were once trafficked into a target country are re-trafficked within that country's borders to other forms of labor, i.e., from indentured labor on the high seas to paid labor in factories (Brayley and Cockbain 2014, p. 172; Sandar 2011).

The Ministry of Labor and its inspection team have worked in Samut Sakhon province to prevent labor exploitation against migrant workers in fishing vessels and the fishery sector since 2014 (SS, personal communication, 2 July 2019). The operation of this Ministry has required serious collaboration with other Thai government agencies such as the Royal Thai Navy, the Immigration Bureau, the Department of Fisheries, the National Operation Centre on Prevention and Suppression of Human Trafficking, and the Ministry of Social Development and Human Security in battling forced labor exploitation (SS, personal communication, 2 July 2019). Consequently, Samut Sakhon province has become an anti-forced labor exploitation model province of Thailand because of these government agencies' rigid operations (SS, personal communication, 2 July 2019).

The Thai government has modified and codified some national laws to combat trafficking in persons, including forced labor exploitation and other forms of internal trafficking. (See Stephens (2017) and Sorajjakool (2013) for further discussion on important issues facing the Thai Government.) The government has also established some relevant mechanisms or cooperative programs set together by the Ministry of Labor, the Ministry of Social Development, and the Human Security and Ministry of Interior to prevent this crime (a lawyer from a government agency, personal communication, 2 July 2019). However, there have been no reports revealing the results of such laws and mechanisms in preventing technology-facilitated labor exploitation and internal trafficking. As a result, this research project first argues that though government and non-government agencies have mimicked the basic IT-based technologies and media abused by traffickers to battle labor exploitation, human trafficking perpetrators have utilized IT-based technology and media to facilitate forced labor exploitation in more efficient ways. Secondly, the Thai government has launched some technologies to prevent technology-facilitated forced labor exploitation through some information or public announcement channels (mobile phone- or computer-based social media or other communication media). Nevertheless, this government prevention measure remains ineffective in preventing such exploitation because of limited information and communication gaps. As the Network Approach posits, information asymmetries and communication gaps lead to an imbalance in power relations between those with information and knowledge and those without (Latonero et al. 2015, p. 29). Using examples from recruitment processes in the Philippines, Latonero et al. (2015) showed how migrant workers without access to the same information as recruiters and employers can experience some negative consequences, including potential indicators of trafficking such as the imposition of high fees, debt bondage, loan-sharking, contract swapping, and social isolation.

### 1.1. Research Questions

This paper asks the following research questions. First, what are the preventive approaches of technology to reduce technology-facilitated labor exploitation through Internal Trafficking in Samut Sakhon, Thailand? Second, what are the results of such approaches? These research questions cater to these two research objectives, i.e., (1) to investigate the preventive approaches of technology to prevent technology-facilitated labor exploitation in Samut Sakhon province, Thailand and (2) to reveal the results or lack thereof of such approaches for the possible improvement of such methods.

### 1.2. Literature Review

This literature review revisits a variety of articles relating to this research project. It includes some terminologies around the phenomenon under study, the general roles and impacts of the internet and technologies in trafficking and information asymmetries, and communication gaps at the heart of trafficking. These articles provide initial responses to the role and impact of technologies on forced labor exploitation. The literature review will cover (1) terminologies, (2) the roles/impact of IT-based technology or media, and (3) information asymmetries and communication gaps.

### 1.3. Terminologies

One of the most apparent distinctions between human smuggling and human trafficking is given by Leman and Janssens (2007, p. 1379) through the use of the United Nations protocols. The first UN protocol (against the Smuggling of Migrants by Land, Sea, and Air) (UN 2000a, p. 3) states that "Smuggling of migrants shall mean the procurement, in order to obtain, directly or indirectly, a financial or other material benefits, of the illegal entry of a person into a State Party of which the person is not a national or permanent resident." The second UN protocol (UN 2000b, p. 3) defines trafficking as "Exploitation, as being part of trafficking shall include sexual exploitation, forced labor or services, slavery or practices similar to slavery, servitude or the removal of organs." Despite this attempt to differentiate between human smuggling and trafficking, Meese et al. (1998) reported that several studies have revealed the quick degeneration of human smuggling into human trafficking. Some Australian scholars, such as Missbach (2015), consistently follow this distinction in their analyses of human smuggling to Australia by emphasizing the aspect of "procurement . . . to obtain . . . a financial or other material benefits" in the phenomenon of people-smuggling. The Thai government has had some historical and economic reasons for not addressing the migrant labor issues in Samut Sakhon (or Mahachai) for many years (Areeprachakun 2020, pp. 10–13).

### 1.4. Roles and Impacts of Internet and Technologies in Trafficking

The internet is or can be used in job recruitment through (1) a recruitment process that includes advertising job offers, advertising job seekers, or the provision of advice for a candidate; the internet can also be a tool to manage the application process and (2) public relations and the legitimation of job offers; a professionally set website legitimizes these business and job offers (FINE TUNE 2011). The internet is a cheap place to post fraudulent job offers for those who look for jobs. The internet is an excellent place of anonymity. It is helpful for communicating with candidate victims by email or chats (FINE TUNE 2011). With internet access in Europe reaching 70.5 percent of the population as of June 2014, most people over a certain age have access to the internet. However, not all current European victims of trafficking were recruited or exploited through online applications. The trend, however, is changing. In November 2014, Europol published a finding that "the Internet is a key facilitator for Trafficking of Human Beings (THB) with an impact on the entire trafficking chain from recruitment and transportation to the harboring of the victims and exploitation" (FINE TUNE 2011, p. 10). A prominent case of trafficking for labor exploitation through internet facilitation was the case of Romanian workers trafficked to Cyprus (FINE TUNE 2011, p. 11). A group of Cypriots and four Romanians (FINE TUNE

2011) established five companies to select and recruit workers through the advertisement of jobs abroad on the internet and in some newspapers with a promised income of between 800 and 1,400 Euros a month (free lodging, free meal, and 8-hour working a day for five days a week). Once they arrived in Cyprus, the recruiters seized the workers' passports and contracts and then sent them to different employers to work for around 16 hours daily in dirty accommodations (FINE TUNE 2011, p. 11).

Human trafficking perpetrators nowadays recruit victims and attract customers through IT-based technologies, namely mobile phones, the internet, and online social networks (Sykiotou 2017). Technology can also be "a means" to strengthen anti-human trafficking strategies (Latonero et al. 2015). In a study in the Philippines, Latonero et al. (2015, p. 22) reported a hypothesis that in both formal and informal recruitment, an imbalance in information appears between job seekers and other relevant actors. For instance, a labor trafficking victim may know very little about the realities of the work and its conditions ahead of time. In contrast, various other actors in the trafficking network might possess and conceal this knowledge. Thailand has already produced at least two laws against human trafficking (Anti-Human Trafficking Act, B.E 2551/2008 published in the Government Gazette Vol. 125, Part 29a, page 28, dated 6 February 2551/2008) (Center for Translation and Language Services, RILCA, Mahidol University 2008) and against cyber-attacks or abuses toward public order or public security (Cybersecurity Act B.E 2562/2019) (The Thai Government 2019).

### 1.5. Information Asymmetries and Communication Gaps in Trafficking

Latonero's Network Approach to understanding recruitment and migration risks posits that the recruitment process embodies a complex network of actors, points of connectivity, information, and communication flows (Latonero et al. 2015, p. 24). As the initial entry into understanding how technological platforms such as Facebook or recruitment websites mediate these connections (before, during, and after the search for jobs), the Network Approach assumes that "at every stage of the migration process, there is a risk" (Latonero et al. 2015, p. 22). From the Philippines field research, this Network Approach hypothesizes that official channels that provide avenues and opportunities for communication between migrants and the web of actors might have the potential to mitigate risk for migrant workers (Latonero et al. 2015, p. 22).

### 1.6. Conceptual Framework

This research applies the concept of "technology-facilitated trafficking" and "internal trafficking" (see Figure 1: Technology-facilitated forced labor exploitation and limited information and communication gaps) to illustrate the limited information and the communication gaps between the government and non-government agencies and migrant workers (International Centre for Missing and Exploited Children 2018; Taylor and Shih 2019). The term "mobile-phone-based technologies" refers to the services of phone calls, smartphone apps (namely the Line application, WhatsApp, Instagram, and so on), social media networks such as Facebook, Google, etc., and email services (such as Gmail, Hotmail, and others) (Taylor and Shih 2019).

These gaps have caused vulnerabilities in migrant workers because of limited information and communication gaps (Latonero's Network Approach). The flow of information has become a constitutive force of society (Castells 1998, 2005). Such a network society also turns to electronic and digital ITCs in their central roles in society by altering individual users' sense of time and space (Latonero 2012, p. 8). The research team agree with Qiu (2009) that network technologies, especially smartphones, have greatly influenced migrant workers' migration plans. In addition, Latonero (2012) argued that these technologies could include some and exclude others in specific ways. For instance, technology can be experienced or accessed differently according to gender, nationality, educational attainment, or skill levels. Migrant domestic or factory workers (most likely women) can easily experience isolation from social support networks and communication channels (Castells 2009). The

older, traditional institutions such as states and governments face tension with the power of global or national networks, which share information, data, and ideas across borders and boundaries at a higher speed and density (Latonero et al. 2015). As a result, government and non-government agencies must be aware of numerous international criminal activities and the dynamics of the supply and demand of inter-border trafficking (Latonero 2012, p. 9; Castells 1998).

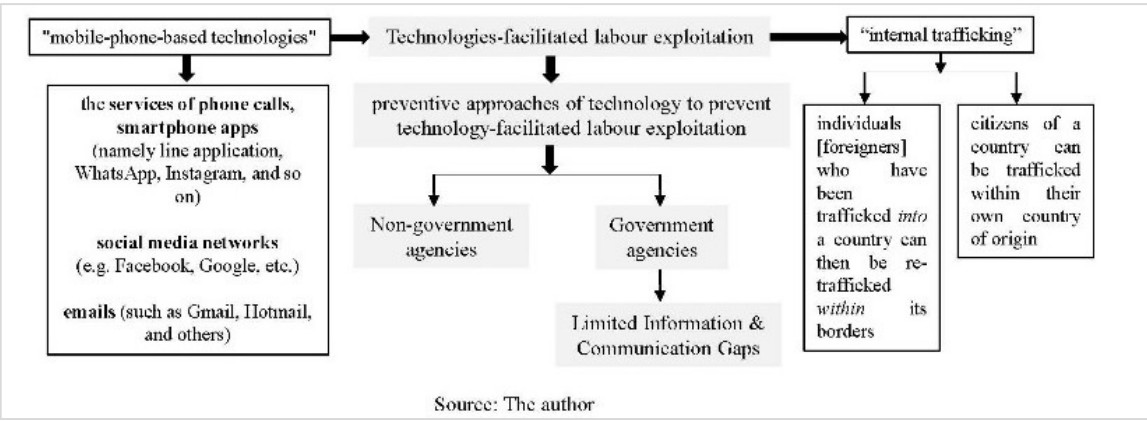

**Figure 1.** Technology-facilitated forced labor exploitation and limited information and communication gaps.

Internal trafficking is a concept introduced by Brayley and Cockbain (2014) to clear up the confusion around the question of whether British children could be victimized in England's territory. The introduction of the legal concept of "internal trafficking" produced clarity on how this crime can be distinguished from other forms of "inter-border" or international trafficking or exploitation (Brayley and Cockbain 2014, p. 171). When the issue of victims' nationality becomes contentious, these authors divide the phenomenon of "internal trafficking" into two forms. First, individuals (foreigners) who have been trafficked *into* a country can then be re-trafficked *within* the country's borders (Brayley and Cockbain 2014, p. 172). Second, citizens of a country can be trafficked within their own country of origin (Brayley and Cockbain 2014, p. 173).

This concept also shows that the traditional Thai state and governance structures face highly significant challenges in policing a transnational crime that is technologically mediated or facilitated and an effect of macroeconomic forces (in internal trafficking), as referred to in the research of Latonero (2012, p. 9). When governments control visas and immigration quotas to regulate national labor markets and international migration, many factors might "push" individuals to circumvent these regulations and seek informal networks and irregular routes to employment. Such circumventions potentially isolate individuals from families, government agencies, services, and human rights advocates, thus increasing their vulnerability to exploitation (Latonero 2012, p. 9). The lack of rights for migrant workers in destination countries also forces them to work under exploitative conditions and possibly numerous venues or factory sites. Therefore, it is more likely that exploitative recruiters or other agencies might exploit such vulnerabilities when one party has more knowledge or access to limited information.

## 2. Materials and Methods

This research aims to answer two research questions, namely, (1) What are the government's preventive and combative approaches through IT-based technology and media to reduce technology-facilitated labor exploitation through internal trafficking in Samut Sakhon, Thailand? and (2) What are the results of such approaches? This research utilizes both primary and secondary data. The research team reviewed documents/reports (of international, regional, national, and local organizations as well as intergovernmental

organizations such as the United Nations and others related to communication technologies and human trafficking) to study the findings of technology-facilitated trafficking in persons, preventive approaches against human (internal) trafficking, and other core facts. Subsequently, the research team interviewed concerned stakeholders, such as NGO staff and government authorities.

The study population and selection rationale (see Table 1: List of Participants) were set as follows. Two key informants from non-governmental organizations have researched and worked on the labor exploitation phenomenon in Samut Sakhon. First, Mr. Sompong Srakaew[1] (henceforth, SS) from the Labor Protection Network (LPN) has offered preventive approaches through information and communication technology—the same ones used by traffickers or internal traffickers—to prevent or reduce human trafficking in Samut Sakhon. SS has also been considered one of the most credible and reliable activists for migrant workers from Myanmar, Cambodia, and Laos. Second, Dr. Lisa Rende Taylor (afterward, LT) from the Issara Institute formerly worked at UNIAP (the present office is called "UNACT") for many years before the establishment of the Issara Institute. LT has recently researched and published a paper entitled "Worker Feedback Technologies and Combatting Modern Slavery in Global Supply Chains: Examining the Effectiveness of Remediation-Oriented and Due-Diligence-Oriented Technologies in identifying and Addressing Forced Labor and Human Trafficking" in the Journal of the British Academy.

**Table 1.** List of Participants.

| No | Name | Organization | Interview Date |
|----|------|-------------|----------------|
| 1 | Mr. Sompong Srakaew | Labor Protection Network | 2 July 2019 |
| 2 | Dr. Lisa R Taylor | Issara Institute | 10 July 2019 |
| 3 | Representative #1 (anonymous) | Min. of Social Development Human Security (MEDHS) | 28 June 2019 |
| 4 | Representative #2 (anonymous) | Min. of Social Development and Human Security (MEDHS) | 1 July 2019 |
| 5 | Representative #1 (anonymous) | Anti-trafficking in Persons Ministry of Interior (MI) | 25 June 2019 |
| 6 | Representative #2 (anonymous) | Anti-trafficking in Persons Ministry of Interior (MI) | 25 June 2019 |
| 7 | Police Lt. Col. Supat Thamthanarug | Department of Special Investigation | 20 June 2019 |
| 8 | Mr. Nakhon Wangphiboon | Min. of Labor Protection and Labor (MLPW) | 4 July 2019 |
| 9 | A representative (anonymous) | Dept. of Fisheries, Port In and Por Out, Ministry of Agriculture and Cooperatives | 4 July 2019 |
| 10 | A representative (anonymous) | Samul Sakhon Governor Office | 17 June 2019 |
| 11 | A policymaker (anonymous) | A government agency | 2 July 2019 |

The interview questions were divided into sections. The first section involves the NGO staff on the role of IT-based technologies and media in anti-human trafficking campaigns and how NGOs have been involved in preventive or other measures against this ubiquitous crime. The interview items for government authorities cover the correlation between information and communication technology, forced labor exploitation, and the existing national laws concerning labor exploitation. Moreover, the interview items include the role and effect of information and communication technologies in human trafficking in Thailand.

The study site is Samut Sakhon because this province houses around 350,000 to 400,000 migrant workers using information and communication technologies that may be subjected to technology-facilitated labor exploitation and internal trafficking. Only 169,230 documented workers from Myanmar work in Samut Sakhon Province (87,872 male workers and 81,358 female workers) (SS, personal communication, 2 July 2019). However, government officials and NGO officials estimated around 400,000 documented and undocumented migrant workers from Myanmar in Samut Sakhon Province. Thus, Samut Sakhon Province has become home to the largest migrant Burmese community in

Thailand, excluding the Bangkok metropolitan area. Major non-government organizations such as the LPN and the Issara Institute have designed and offered the same information and communication technologies used by traffickers to prevent or reduce human trafficking or internal trafficking. Moreover, national government agencies have closely collaborated with the LPN to raise awareness of forced labor exploitation. The research team conducted this research between June and July 2019.

## 3. Results and Discussion

This section reveals two issues. The first issue includes the role and effect of technologies in preventing human trafficking, the current anti-trafficking laws, and the challenges of searching for methods to prevent technology-facilitated inter-border and internal trafficking. These findings reflect how technologies might prevent forced labor exploitation in Thailand by mimicking the modus operandi of the traffickers. The second matter refers to the results of the preventive approaches of technologies to fight against human trafficking. These results disclose the current preventive approaches of the technology of Thai government agencies and non-government organizations in Thailand to suppress technology-facilitated forced labor exploitation.

### 3.1. The Role and Impact of Technologies in Human Trafficking

Regarding the role of technologies, LT from the Issara Institute (personal communication, 10 July 2019) explained that mobile technologies had influenced Burmese migrant workers in the recruitment process. Technology can potentially build migrant workers' power and organizations, especially in countries that do not give room for freedom of association. Remarkably, there are cases of foreign migrant workers who cannot form trade unions, such as in Thailand and Malaysia. Facebook groups and other apps have been created to help form online spaces for workers to support each other, exchange experiences, and organize negotiations with their employers. LT explained the following:

> " . . . All of a sudden, there were millions of Burmese people on Facebook, Viber, etc. So, one of the first big advantages of technology is its ability to scale up messaging and assistance to workers in a highly cost-effective manner. For example, in shorter than two years, we got over 150,000 Burmese followers on our Issara Myanmar-language Facebook". (LT, personal communication, 10 July 2019)

Regarding the positive role of technologies, Supat Thamthanarug (henceforth, ST) from the DSI (personal communication, 20 June 2019) shared that technologies help disseminate information about cases and laws to educate people, including business owners. For example, they can inform people that confiscating a passport is considered a deprivation of freedom of movement. However, business owners can still keep their employees' passports if employees provide consent. The authorities can also warn business owners/employers not to exploit employees while raising awareness of the potential and current employees with human trafficking elements. Laborers use a cellphone to inform their relatives or to call for help from the DSI. They can call a central phone number to share their complaints (ST, personal communication, 20 June 2019).

Regarding the negative role and effect of technologies, all participants agreed that technologies also serve as vital tools for human traffickers to deceive victims. For instance, both sources (R#1 of the MSDHS, personal communication, 28 June 2019, and R#3 of MI, personal communication, 25 June 2019) agreed that technologies cause or facilitate human trafficking. Perpetrators increasingly announce job opportunities on the internet, on Facebook, in inbox messages, or on the Line application, which allows them to attract workers. One source (R#4 of the MI, personal communication, 25 June 2019) explained that "Perpetrators use Facebook to post fake news on job opportunities." Another source (R#2 of the MSDHS, personal communication, 1 July 2019) informed us that "Technologies allow perpetrators to attract victims through their advertisement. Perpetrators can contact or access vulnerable people or victims more easily via the various channels of IT-based technologies". Similarly, LT (personal communication, 10 July 2019) shared that the negative impact of technologies

on such migrant workers is that "job advertisements are spread by brokers through the Burma countryside now through Facebook more than through in-person contacts." Technologies, therefore, could play an essential role in facilitating human trafficking. However, perpetrators might also employ IT-based technologies (websites, online applications, etc.) to attract victims to sex trafficking, just like forced labor exploitation, in Samut Sakhon, Thailand.

*3.2. The Current Trafficking Laws and Implementation*

The statements of the most concerned participants (Representatives R#1 and R#2 of MSDHS, Nakhon Wangphiboon (NW) from the MLPW, and ST from the DSI, personal communication, 28 June, 20 June, and 4 July 2019, respectively) acknowledged that the Thai national laws related to anti-human trafficking facilitated by technologies mainly include a long list of regulations and acts. The sources (R#1 and R#2 of the MSDHS, personal communication, 28 June 2019) reported that these national laws aim to prevent human trafficking, protect trafficking victims, and prosecute perpetrators. Law practitioners are empowered to apply these laws to all human trafficking cases, including technology-facilitated forced labor exploitation cases.

The findings illustrate that law practitioners have utilized IT-based technologies and media to search for core pieces of evidence to prove the misconduct of perpetrators. These technologies have served as powerful instruments to issue complaints of forced labor exploitation. A government source (ST from the DSI, personal communication, 20 June 2019) stated that victims typically tell the story of technology-facilitated trafficking by showing their cell phones as evidence. Migrant workers use cell phones to take photos of vessels or work venues. For instance, photos can prove the working time someone has gone through in child labor cases to prove one's age (ST from the DSI, personal communication, 20 June 2019). Moreover, he added that "[through] the line application [it] is hard to get personal contact because of the big size of the group membership. The system requires much time just to read messages. It has too many messages and takes too much battery power. Facebook is better because the members can do activities at any time. There is no hotline for 24 hours, but office hours are given with interpreters". The fact is that all types of technologies can create language barriers. Technologies are tools to educate only (ST from the DSI, personal communication, 20 June 2019). Likewise, the source (R#1 of the MSDHS, personal communication, 28 June 2019) clarified that the Ministry would protect victims such as children more than just process them through victim identification processes. The officers are not searching for evidence. However, victims may share information with staff about how technology led them to forced labor exploitation. Then, the staff reports the situation to police (R#1 of MSDHS, personal communication, 28 June 2019).

Most government agencies and NGOs have used technologies to prevent forced labor exploitation or internal trafficking. ST from the DSI (personal communication, 20 June 2019) informed us that the DSI has collaborated with numerous ASEAN countries' governments and agents to share evidence with the DSI. The DSI has collaborated with the Embassy of Myanmar to monitor migrant workers' workplaces. ST (personal communication, 20 June 2019) explained that migrant workers in Thailand could contact other migrant workers on a Facebook group that shares the Embassy's phone. Subsequently, the Embassy will send information to them to identify whether they are dealing with human trafficking. Primarily, the Embassy uses cell phones and messenger groups on Facebook for public communications. Additionally, non-government organizations have been involved in different processes such as interpretation and translation services, legal assistance, etc., because victims trust NGOs more than other government enforcers.

The DSI (R#1 and R#2 of the MSDHS, personal communication, 28 June 2019) created its mobile application named "ProtectU" (U means "you"). People can simply start using Android/IOS systems (iPhone or others in March 2019 to open a new channel for victims to report a case), and victims can ask for help aside from calling the police. Typewriting and addressing their location are possible, but with Thai alphabets only. The MSDHS is

planning to add more languages, including English. Nevertheless, the MSDHS may also soon develop more applications with other languages. ProtectU has been advertised in all events and websites, passport centers, and learning centers of MSDHS and at other protection centers such as the "Kietrakarn Centre." However, ProtectU has no reports of victims of forced labor exploitation because it has just been newly installed. Victims tend to use the phone (#1300 is the phone number of the Division of Social assistance that receives all complaints of social problems, including forced labor exploitation situations) more than other IT- or technology-based tools (R#1 and R#2 of the MSDHS, personal communication, 28 June 28 and 1 July 2019).

Government sources (R#3 and R#4 of MI, personal communication, 25 June 2019) mentioned that the MI created the "Ta-Thip" application in a Thai version to search thoroughly for online sex trafficking, but not forced labor exploitation. Further, the MI uses its website and Facebook to disseminate and advertise information on court cases and awareness-raising campaigns on general human trafficking categories and issues. Another channel through which victims can contact the MI is the phone number 1191, which may not fully serve them because of the constraint of the number of interpreters (R#3 of the MI, personal communication, 25 June 2019).

A government source (NW from the MLPW, personal communication, 4 July 2019) specified that its call center number 1506 serves as a "Complaint Channel" to prevent all forms of human trafficking (including forced labor exploitation), unfair labor contracts, illegal migration, registration of migrant workers, etc. It offers English, Myanmar, and Cambodian translation/interpretation on its own. It also created a website (www.doehelpme.org accessed on 27 October 2021) through which victims can file a report. Moreover, the MI has worked closely with other government agencies. However, the MI has no resources for calling to check on migrant workers. In addition, there is no technology integration to formulate a central information database among government agencies because each Ministry has its own technology use. The MI has fewer human resources and a low budget than its workload (R#3 of the MI, personal communication, 4 July 2019).

A representative from the Department of Fisheries, Port in and Port out Control Center (PIPO), Ministry of Agriculture and Cooperatives explained that PIPO designed a 24-hour vessel-monitoring system (VMS), including illegal, unreported, and unregulated (IUU) fishing, to check documents related to vessels, seamen, proper operations, and treatment on vessels. However, PIPO rarely uses technologies in the monitoring process but rather in the individual check approach (R#5 of the Department of Fishery, personal communication, 4 July 2019).

It is imperative to note that although the number of technology users is expanding, the Thai government agencies may not be sufficiently prepared for the safe migration of laborers. For instance, NW from the MLPW (personal communication, 28 June 2019) described that technology use in the MLPW is still low. However, the Ministry has endeavored to use its website to campaign against human trafficking and inform users of the news of forced labor exploitation (NW of the MLPW, personal communication, 4 July 2019). R#1 of the MSDHS (personal communication, 28 June 2019) commented that government agencies use technologies to report human trafficking rather than only victims reporting to government agencies. The MLPW designed its mobile application named "ProtectU" which aims not only to report human trafficking cases but also to encourage others to buy products produced by victims to help them earn income. This idea to support victims came from Lazada. A source (a policymaker of a government agency, personal communication, 2 July 2019) reflected that "Technology use among the government agents will help the practitioners understand the whole process of a human trafficking situation."

Similar to other government agencies, SS from the Labor Protection Network (LPN) (personal communication, 2 July 2019) agreed that various migrant workers prefer to share their complaints of human trafficking situations on the phone or on the cell phone of other staff, including interpreters/translators, or on Facebook, or specific Facebook groups of Myanmar, Cambodia, and Laos. There is a Myanmar migrant labor group (MMLG), a

Cambodia migrant labor group (CBLG), and a Laos migrant labor group (LLG) that aim to recruit people to update information for each group. The LPN also has an online live stream of its activities—training or events on all issues to advertising the LPN. Otherwise, they report their concerns to the Thai staff if they can speak Thai. However, they barely contact the staff via the Line application because of the language barrier (SS, personal communication, 2 July 2019).

Serving an NGO such as the LPN, LT from the Issara Institute shared the following information in this issue:

> "Our main technologies are Golden Dreams, plus the other worker voice channels—Line and Viber, Facebook, and the 'traditional' 24-hour freephone helpline. The first important thing that we do is have a range of different channels according to the preferences of different workers–some workers like to chat with us and share their concerns over Viber . . . others like to hear a human being on the other end of a phone. Others like to research Golden Dreams, reading the reviews of thousands of other Burmese workers to learn which employers and recruiters are the best—and even NGOs and service providers like us. . . . we try to get the employer to address the issue". (LT, personal communication, 10 July 2019)

LT (personal communication, 2 July 2019) also added the vital point that "job seekers and workers are less likely to be trafficked or exploited if they (1) are empowered with specific, updated information about how to find a good job, know their rights, what to do if anything goes wrong (effective hotlines, etc.) and (2) if workers can grow their collective power in their negotiations with their employers".

Many technology users and victims have encountered limitations of each technology (cellphones, websites, and social media such as Facebook or other applications) because they need interpreters or translations. A source (a policymaker of a government agency, personal communication, 2 July 2019) shared that technologies are perceived as all good; however, the most important thing is the effectiveness of these tools, access to information, etc. Therefore, technologies will be helpful if Thai government agencies and non-government organizations provide interpretation and translation services.

ST from the DSI (personal communication, 20 June 2019) admitted that the DSI had designed its website for awareness-raising purposes. However, few migrant workers can access it because of the language barrier. Yet there is no information about the possible risks of human trafficking. Collaboration with the governments and NGOs of Myanmar, Laos, Cambodia, Vietnam, Malaysia, Indonesia, Philippines, etc. is carried out to share information. NGOs also help victims before the DSI arrives because victims will trust them more than the authorities.

Moreover, the DSI website does not address other significant information such as people or workplaces that can be a risk for workers (ST from the DSI, personal communication, 20 June 2019). Similarly, a representative from the Samut Sakhon Governor's Office shared that the Samut Sakhon Governor's Office has not officially applied technologies to the fight against human trafficking. Its authorities have collaborated with the Internal Security Operations Command staff only to investigate the data on the issue. Samut Sakhon province is not a "smart city yet" (R#6 of Samut Sakhon's Governor Office, personal communication, 17 June 2019).

A government source (R#6 from the Department of Fisheries, Port In and Port Out Control Center or PIPO, Ministry of Agriculture and Cooperatives, personal communication, 4 July 2019) introduced us to the fact that the "PIPO (Ministry of Labor) requires all migrant workers to scan their fingers, but this system is practical only for legal migrant workers." The number of those registering/scanning their information is only about 200,000 to 300,000 legal migrant workers. PIPO focuses only on fishing because of the IUU mandate, but not other forms of human trafficking, namely construction, working in industries, etc. (R#6 from the Department of Fisheries, Port In and Port Out Control Center (PIPO), personal communication, 4 July 2019).

ST from the DSI (personal communication, 20 June 2019) raised concerns that the application of the network may not function to prevent human trafficking because not everyone knows the name of the application. The website of the DSI announced that anyone could contact the DSI through Dropbox (email). Besides watching perpetrators' websites, the DSI and police have positioned their teams to monitor other possible dark websites, such as pornographic websites. The DSI does not have an application, but the National Police Office has designed an "I love U application" to warn or receive human trafficking information. The DSI mostly received complaints through emails or paper communication (ST, personal communication, 20 June 2019).

Another source (R#1 of the MSDHS, personal communication, 28 June 2019) commented that technology is an innovation. Nevertheless, the challenge is how to allow victims access to technological tools. The budget is not a big issue because the government has allocated it annually (R#1 of the MSDHS, personal communication, 28 June 2019). The other source from the MSDHS (Representative R#2 of MSDHS, personal communication, 1 July 2019) added that the challenges are the new forms of human trafficking; thus, government authorities need to study them more to prevent misconduct. The MSDHS aims to give information on preventing children from being trafficked through anti-human trafficking in children's tool kits by visiting schools to inform children and parents in every province (twice a year for every province) (R#2 of the MSDHS, personal communication, 1 July 2019).

A source (R#1 of the MI, personal communication, 25 June 2019) discussed that the MI has tried to share general information and news on general cases on their website to warn people about the human trafficking situation. There is no challenge to the prevention approach because the MI believes its website offers the best channel for news distribution, court cases, human trafficking situations, etc. (R#3 of the MI, personal communication, 25 June 2019).

A policymaker of a government agency complained that the government tends to classify the type of migrants more than prevent human trafficking such as forced labor exploitation. The established mechanisms have been formulated to respond to international community pressures, and the government has not initiated mechanisms by itself (anonymous policymaker, personal communication, 2 July 2019).

## 4. Conclusions

Technologies might have played a significant role in preventing and reducing human trafficking and internal trafficking through different communication channels between government and non-government agencies and migrant workers. NGO campaigns using native speakers or former victims appear to produce better results than those of the government's campaigns using mainly IT technology in reaching out to potential and actual victims of trafficking. However, such channels have produced loopholes and insufficiency of information and communications. Therefore, these prevention approaches, primarily as provided by government agencies, need to be improved to prevent labor exploitation.

### 4.1. IT-Based Technologies and Media as Combat Tools

Based on the findings, there are two communication channels: between government agents and potential migrant workers and between non-government organizations (NGOs) or migrant worker groups and potential migrant workers. As government and non-government organizations have applied technologies to prevent labor exploitation, perpetrators have employed various technology tools/channels to promote their crimes.

In combatting forced labor exploitation, the information and communication channels between the non-government organizations or migrant worker groups and potential migrant workers have been considered peer-to-peer networks and vital sources of anti-forced labor exploitation. SS from the LPN and LT from the Issara Institute (Issara Institute and IJM 2017) have strongly confirmed that before their arrival in Samut Sakhon province in Thailand, many potential migrant workers in Myanmar, Cambodia, and Laos preferred to contact NGOs or migrant workers groups via cellphone, websites, and online applications

to inquire about safe migration. Compared to the government authorities, those NGOs and migrant worker groups are certainly more approachable and reliable for these potential and existing migrant workers. The work of Latonero et al. (2015), for example, also showed that peer-to-peer networks were crucial sources of information and communication for migrant workers. Migrant workers from Myanmar, Cambodia, and Laos have chosen a "trusted" recruitment agency to migrate to certain countries. Facebook has enabled informal recruitment; some NGO staff reported that unaccredited recruiters had contacted some beneficiaries (migrant workers) via Facebook. According to some social workers, Facebook contacts usually solicit phone numbers and set a series of conversations to establish trust and enforce a decision to migrate, such as the findings of Latonero et al. reveal (2015, pp. 28–29). However, during the points of connection, formal and informal recruitment agencies utilize many means of recruitment, from low-tech fliers to Facebook chats. The workers, who are job seekers on the informal channels, may experience expedited processes. However, these job seekers are potentially disconnected from the verified information and support networks that are more accredited (Latonero et al. 2015, p. 23).

This paper has found that preventing forced labor exploitation is possible through IT-based technology and media or applications. Though the NGOs and migrant worker groups have been sources of important information and communication resources for potential migrant workers in Samut Sakhon province, human trafficking networks and internal trafficking have utilized direct connections with migrant job seekers and numerous high technologies (namely cellphones, Facebook, Instagram, and other online applications) to deceive migrant job seekers who plan to work in Thailand. These perpetrators have attracted migrant job seekers by disseminating fake news and bogus advertisements of job descriptions.

### 4.2. Results of the Prevention Approaches

The non-government agencies and migrant worker groups, on the one hand, have presented a strong performance in information dissemination and interaction with potential migrant workers to raise awareness of labor exploitation as prevention measures. On the other hand, government agencies have struggled with budgetary and human resource constraints.

Without online live materials, such attempts to "prevent" trafficking for labor are unlikely to produce any noticeable results (SS, personal communication, 2 July 2019). The most logical way to create a prevention or post-trafficking solution campaign is to disclose the unfavorable events when every node of connectivity has loopholes for imbalanced power relations (and thus potential exploitation). The secret of success by the Labor Protection Network of SS is in the application of "remedies" to rehabilitate the yawning gap in both "communication gaps" and "information asymmetries" that already take victims through vertically set nodes of connectivity (between job seekers and recruiters) (SS, personal communication, 2 July 2019). The LPN's "horizontally-set" (through mother-tongue "peers") communication bridge to rehabilitate the "information asymmetries" produced by recruiters and "communication gaps" (either through the confiscation of mobile phones or the provision of the phone as the only source of assistance to job seekers) has worked wonders to restore the imbalanced power relations/information asymmetries that put job seekers in social or physical isolation. This analysis reveals the results of some initiatives in gaining certain levels of success, minimal success, or no success at all.

However, a lack of equal access to the same information set as recruiters and potential employers might bring some negative experiences to government agencies (Latonero et al. 2015, p. 29). Consequently, building the skills of migrant workers to use technology and the internet to verify the information and seek assistance in times of distress can counter the misinformation disseminated through social media, the internet, and low-tech tools (Latonero et al. 2015, p. 33).

The efforts of the Ministries of Labor, Internal Security, Social Development, and others in "preventing" or "regulating" these labor practices face significant obstacles, i.e., a lack of

trust from the labor forces in the government's truthfulness to work for laborers' interests. No attempt has been made to create a proper first stage of the "right connection" between labor and officials and trusted recruiters. Due to the lack of sufficient information and the communication gaps between the Thai government and its agencies, these agencies should be aware that brokers could simply chat with victims/migrant workers or speak their language to overcome the language barriers in online channels. As a result, government agencies should be more proactive in their prevention approaches through their websites, hotline phone numbers, and other social media networks that migrants can easily access to find assistance. It should encourage the government to continue doing its work of enforcement and regulation. It is also essential for governments to embrace conversations and collaboration with NGOs and migrant worker groups to provide sufficient information and efficient communication tools and channels for potential migrant workers or migrant job seekers. For instance, the Thai Government could cooperate with NGOs with access to native speakers or experienced staff who are previous victims of human trafficking to improve the materials or communication lines to reach out to the potential or actual victims of "internal trafficking" or "inter-border" trafficking and other types of trafficking in Thailand.

**Author Contributions:** Conceptualization, N.K. and Y.S.; methodology, N.K.; formal analysis, N.K. and Y.S.; investigation, N.K.; resources, N.K.; original draft preparation, N.K.; writing, N.K. and Y.S.; writing—review and editing, N.K. and Y.S.; visualization, N.K. All authors have read and agreed to the published version of the manuscript.

**Funding:** This research received no external funding.

**Institutional Review Board Statement:** The study was conducted in accordance with the Declaration of Helsinki, and approved by the Institutional Review Board of the Institute for Population and Social Research, Mahidol University (Protocol code 2019/04-124 and date of approval 25 April 2019).

**Informed Consent Statement:** Informed consent was obtained from all subjects involved in the study.

**Conflicts of Interest:** The authors declare no conflict of interest.

## Note

1. This participant and others understand that they are offered anonymity in the final research report; nevertheless, those with real identities agreed that their names or other identities are revealed because no security or ethical concerns are perceived.

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
