# Peer review of "Failed Mimicry: The Thai Government’s Attempts to Combat Labor Trafficking Using Perpetrators’ Means"

_socsci, doi:10.3390/socsci11090422_

Round 1

Reviewer 1 Report

This manuscript addresses an important human rights issue. The manuscript is generally well written and well documented. The literature review is up to date.  I learned a lot about how labor traffickers are effectively using IT-based technologies to recruit victims.

I found the author’s analysis of the three types of trafficking to be thought-provoking in a good way:

The well-known three types of trafficking (for labor, sexual, and organ exploitation) 107 are critical in finding a way to deal with or prevent them. Efrat (2016, p. 34) claimed that 108 the significant differences among the three practices would significantly affect govern-109 ments' willingness and ability to curb them. Israel, for example, has been vigorous in com-110 bating trafficking for sexual exploitation but is hesitant to tackle labor and organ traffick-111 ing (Efrat 2016, p. 34). These differences in commitment are caused by three factors in-112 volved in human trafficking: (1) varieties in the perpetrators' legal and social statuses and 113 thus their parts in policy discussion, (2) varieties in clarity, precision, and resonance (to 114 the audience) of norms against each type of trafficking, and (3) varieties in enforcement 115 costs to deal with each type of trafficking. The Thai government, for example, has some 116 historical and economic reasons for not sensitizing the migrant labor issues in Samut 117 Sakhon (or Mahachai) for many years (Areeprachakun 2020, pp. 10-13).

This discussion made me wonder where trafficking in migrants, sometimes referred to as human smuggling fit in. Since this issue is at the forefront of current international debates about migration, the author might want to add a paragraph to her paper. Is human smuggling a form of human trafficking? Isn’t the boundary between labor trafficking and migrant trafficking blurred? Adding a paragraph on this topic is just a suggestion about a way to further engage potential readers.

From reading this manuscript, I learned a lot about the efforts of the Thai government and Thailand-based NGOs to use the same technological tools that traffickers use to combat trafficking. As I read the paper, the author concludes that those anti-trafficking efforts have not been successful and explains why. The author’s analysis is convincing.

The author summarizes their argument as follows:

Human trafficking perpetrators nowadays recruit victims and attract customers 142 through IT-based technologies, namely mobile phones, the internet, and online social net-143 works (Sykiotou 2017). Technology can also be "a means" to strengthen the anti-human 144 trafficking strategy (Latonero, Wex and Dank 2015).

Unfortunately, as noted, I see little evidence in the article that technology can also be "a means" to strengthen the anti-human trafficking strategy. The author makes a weak argument that NGOs have had more success than government agencies in using technology to combat trafficking (pages 13 and 14). I evaluate the argument as weak because there is no supporting evidence. In a revision, the author could choose to strengthen/clarify this case.

Or the author could contribute by presenting a more detailed case study of how labor traffickers use technology to achieve success. Then, she could explain in greater detail what NGOs are doing to counter those efforts somewhat successfully. Finally, she could explain how the government could collaborate with NGOs to achieve even better results.

 Bottom line: There is limited value in disseminating the results of failed social experiments (“Failed Mimicry”). An article about a successful experiment conducted to reduce labor trafficking may encourage others to adopt similar policies. But I don’t see how policymakers or scholars can build on this work.

Here are some more detailed comments:

The author poses two research questions:

1.1. Research Questions 87

1. What are the preventive approaches of technology to reduce technology-facilitated 88 labor exploitation through Internal Trafficking in Samut Sakhon, Thailand? 89

2. What are the results of such approaches? 90

One suggestion I have is to remove the words “in Samut Sakhon, Thailand” from the first question. If readers are to be interested, the question needs to be stated more broadly. The place name is simply where the authors studied the important question raised.

Maybe the second question should also be rephrased as “Can technology be used to reduce labor exploitation?”

I would also replace the word “exploitation” with “trafficking” in both questions to be more consistent with the previous discussion in the manuscript.

Other Small points:

Some paragraphs are too short. For example, this one on page 3:

Human trafficking perpetrators nowadays recruit victims and attract customers 142 through IT-based technologies, namely mobile phones, the internet, and online social net-143 works (Sykiotou 2017). Technology can also be "a means" to strengthen the anti-human 144 trafficking strategy (Latonero, Wex and Dank 2015).

And these two on page 6:

The interview questions are divided into the following sections. The first section in-249 volves the NGO staff around the role of IT-based technologies and media in anti-human 250 trafficking campaigns and how NGOs have been involved in preventive or other 251 measures against this ubiquitous crime. 252

The interview items for government authorities cover the correlation between infor-253 mation and communication technology, forced labor exploitation, and the existing na-254 tional laws concerning labor exploitation. Moreover, the interview items include the role 255 and effect of information and communication technologies in human trafficking in Thai-256 land.

Author Response

Dear Reviewer,

Here are the responses (with the corresponding improvement in RED ink in the text):

  1. Is human smuggling a form of human trafficking? To this inquiry, the authors added a paragraph on page 2 of 15 (introduction, in red ink) to explain the UN Protocols (against the Smuggling of Migrant ... and to Prevent, Suppress and Punish Trafficking in Person ...) that define the two phenomena. Also added is an example from an Australian scholar who consistently uses human smuggling to cover the illegal migration to Australia (not Trafficking for its 'exploitation' connotation).
  2. Little evidence that technology can also be a means to strengthen anti-human trafficking strategy. To this comment, the authors marked in the text (in orange color) some narratives from NGOs (Labor Protection Network led by Sompong Srakaew) who explain that hiring translators/native speakers of Burmese/Laotian improves reaching the potential/actual victims. These victims become more open and willing to report their plights.
  3. The authors could contribute by presenting a more detailed case study of how traffickers use technology to achieve success. Most information that provoke this research have been informal information shared by NGOs, Government Sources, and victims who noticed the improvisation done by traffickers to ride on IT/Communication technology. Most research in Thailand or Burma rarely studies the perpetrators' methods for high-security concerns. Besides, our attempts to look for this type of study on "how traffickers operate" find more studies on other aspects of trafficking (victims, policy, prevention, punishment). These sources rarely write their findings in scholarly or academic databases. Therefore, the authors humbly would recommend the study on how perpetrators operate in Thailand or Burma for future research.
  4. I don't see how policymakers or scholars can build on this work. To this comment, the authors added the fourth sentence in the Abstract to show the attempt by Labor Protection Network to recruit native speakers/former victims to work in prevention or assistance campaigns among potential/actual victims.
  5. To remove the word "in Samut Sakhon, Thailand from the first question. To this suggestion, the authors believe that removing the province's name would be misleading because the policy is not universally applicable to the other 66 provinces in Thailand. Generalization into "Thailand" would not be precise.
  6. Rephrased second question as "Can technology be used to reduce labor exploitation?" To this suggestion, the same reason like previous suggestion applies. Authors can't generalize "technology" to other technologies because the attempts by government and NGOs to campaign against trafficking are done particularly through internet-related mobile phone applications (not "technology" that has broader connotations).
  7. Replace the word exploitation with "trafficking" in both questions. The UN Protocols (2000) against human smuggling and human trafficking (in two separate supplements to Convention against Transnational Organized Crime) emphasize different concepts for each. The one for anti-smuggling emphasizes "procurement" (for financial benefits), and the one for anti-trafficking emphasizes "exploitation" (sex, labor, forced labor). The word "exploitation" keeps this emphasis better than "trafficking" alone since the phenomenon in Thailand is closely connected to "trafficking" (with connotation strongly of 'exploitation' for some purposes). Thus authors decide to keep the word exploitation as 'trademark' for trafficking.
  8. Some paragraphs are too show. Authors incorporate back these short paragraphs to the relevant, close-by paragraphs to improve the presentation quality of the text. Thank you.

Reviewer 2 Report

This is an interesting discussion of an issue that was not considered two decades ago when the Trafficking Protocol was created and anti-trafficking laws and policies were introduced in a concerted way.  It has strong potential to make a contribution to the literature,

The paper explains how migrant workers from Myanmar who were trafficked onto Thai shipping boats are often then re-trafficked into (mostly) fish processing factories.  It draws attention to the use of technology to recruit workers and ‘argues that while the government and non-government agencies have mimicked the basic IT-based technologies and media abused by traffickers to battle labor exploitation, human trafficking perpetrators have utilized IT-based technology and media to facilitate forced labor exploitation in more efficient ways’ (p2).  It argues that ‘government prevention measure remains ineffective in preventing such exploitation because of limited information and communication gaps’

This is an important argument.  As the authors point out technology can lead to an even greater imbalance in information which can be the basis of an exploitative relationship.

The paper concludes with some suggestions for how the government’s approach could be more effective. 

The paper also includes discussion of a very useful set of interviews.  However, I think that more effective use of them can be made.  For example, explain how they fit under each theme or argument rather than setting them out verbatim.

Importantly it points out that technology is really useful for migrants (from Myanmar?) who cannot join \ form trade unions unlike those in Thailand and Malaysia.

However, the paper and its argument could (should) be much more clearly presented and structured.  It needs to commence with a clear statement of the problem and the argument\ hypothesis.  It includes discussion of some points whose relevance is not entirely clear.  For example, the reference to ‘inter-border’ trafficking and the discussion of 3 types of trafficking …. What is their relevance?  Is the problem limited to those who previously worked on Thai fishing boats or does it also cover those who moved directly from Myanmar?  If the focus is on the fishing boats it would be helpful to explain a bit more about the background and why the Thai government has been so proactive (TIP report?).

In it current format it reads more like a student research paper \ report (setting out research questions at p2 and a literature review).

It might be helpful to have a map to explain the location of Samut Sakhon .

On p7 it raises the issue of whether the current anti-trafficking laws are adequate but on p8 explains they are – it would be much simpler to tell the reader from the start that recruitment via technology is covered by the laws. 

The final recommendations need to be set out more clearly and explained.  For example how does one build the skills of migrant workers to use technology (p13)?  In what ways should government agencies be more proactive? 

Author Response

Dear Reviewer,

Below are our responses (and revisions in the revised text using BLUE ink) to the comments.

  1. The relevance of 'inter-border' trafficking and discussion of three types of trafficking. To this inquiry, the authors believe that many readers are not familiar with human trafficking, and many believe that human trafficking happens between state borders. What these discussions contribute is a general discussion on trafficking and the fact that trafficking can happen more than once for the same victims (Burmese/Lao) within the border of a state (Thailand) from one form of exploitation to the next.
  2. The phenomenon discussed is not just about former fishing boats (just illustrations) labor but also more types of migrants (direct from Burma territory), including surrounding states where labor forces originate (Burma, Lao PDR, Cambodia).
  1. Explain a bit more about the background and reasons for the Thai Govt to be proactive in TIP (Trafficking in Persons) Report. This apparently simple addition to the real focus of study on the failures of government policy (and NGOs) to apply similar methods as used by traffickers would be seriously distractive and prolong the text (with word limits in mind). Therefore, the authors believe that perhaps the best solution is just to mention some sources to understand this issue for interested readers.
  2. It reads more like a student research paper. To this comment, we could only offer our little choice about our status as Mahidol University teachers. Our status as Mahidol staff requires that if we write a "research-based" paper (and want Mahidol committee to adjudicate our papers at such values), we must put explicitly (no matter how naive it might appear) "research questions" and "literature reviews" (also methodology). If we use a different, more indirect style of structure/writing, the possibility that our paper will be dismissed as "normal research" is high, and thus the value of the paper in Mahidol ranks and files would be lower than we've expected.
  3. Provide a map of Samut Sakhon province (as Bangkok's part). This map is technically possible as far as the Journal allows it for space/bytes limitations. Authors can add the province’s map into the text.
  4. Tell the readers from the start that recruitment via technology is covered by the laws. Yes. Thailand has the Anti-Human Trafficking Act B.E 2551/2008 and Cybersecurity Act B.E 2562/2019. Authors might need to mention these laws to show that the Thai Government has laws to protect public order and potential victims of human trafficking (added into the text on page 4 in blue).
  5. How does one builds the skills of migrant workers to use technology? In what ways should government agencies be more proactive? These recommendations would be added into the revised text as appropriate. The addition appears on page 14 in blue ink.

Round 2

Reviewer 1 Report

Sorry, but, despite your revisions to the previous version of the manuscript,  I still do not see how this research report will be useful to other scholars or to policymakers. The research and policy implications are hazy at best. And even if they were clearer, you refuse to generalize your findings to places other than Samut Sakhon province. 

Author Response

Authors thank the reviewer for the comments
